# Testing the effectiveness of a mobile approach avoidance intervention and measuring approach biases in an ecological momentary assessment context: study protocol for a randomised-controlled trial

Matthias Burkard Aulbach  ,[1,2] Hannah van Alebeek,[1,2] Sercan Kahveci,[1,2] Jens Blechert[1,2]

[1]Department of Psychology, Paris Lodron Universitat Salzburg, Salzburg, Austria
[2]Department of Psychology, Centre for Cognitive Neuroscience, Salzburg, Austria

**Correspondence to**
Dr Matthias Burkard Aulbach;
matthias.aulbach@plus.ac.at

## ABSTRACT

**Introduction** Unhealthy eating behaviour is a major contributor to obesity and related diseases and is associated with a behavioural bias to approach rather than avoid desired foods, as measured with reaction time tasks. Approach-avoidance interventions (AAIs) have been proposed as a way to modify food evaluations and help people to eat in accordance with their dietary goals. Mobile implementations of AAI might be easily accessible, low threshold interventions, but their effectiveness has not been established yet.

**Methods and analysis** Participants who aim to change their eating behaviour are randomised to intervention or control groups. They complete six sessions of a smartphone-based AAI, in which they push (ie, avoid) or pull (ie, approach) personalised food images. Intervention group participants always avoid foods that they personally want to eat less often and approach foods that they personally want to eat more often. In the control group, images are paired equally often with both response directions. To evaluate contextual and dynamic intervention effects, ecological momentary assessment (EMA) is measured throughout, with questions about food intake, hunger, stress, emotions, eating intentions, food craving and impulsivity twice a day. Additional EMA preintervention and postintervention measures are administered before and after the intervention phase (4 days each) with a 1-day follow-up EMA 4 weeks after the intervention. Multilevel models will examine the temporal covariance between approach bias and self-reported variables as well as short-term and long-term intervention effects on approach bias, food intake and craving.

**Ethics and dissemination** The study was approved by the Ethics Committee of the University of Salzburg. Results will be published in peer-reviewed scientific journals and presented at scientific conferences.

**Trial registration number** German Clinical Trials Register DRKS, registration number DRKS00030780.

## STRENGTHS AND LIMITATIONS OF THIS STUDY

⇒ The study is a randomised controlled trial testing an m-health intervention to assist participants in implementing their dietary intentions through repeated use of a mobile approach-avoidance intervention compared with a closely matched active control task.

⇒ It includes ecological momentary assessment (EMA) before, during and after the trial in both control and intervention groups, which allows examining both short-term and long-term intervention effects.

⇒ It measures a range of potentially relevant phenomena like food craving, hunger, emotions, stress and day-level impulsivity.

⇒ Control group data allow examining variability in approach bias and its covariation with data obtained through EMA.

⇒ Measures of food intake are restricted to single-item daily self-reports which are prone to under-reporting and experimenter demand.

health, as overeating can lead to obesity and related diseases.[2] It is therefore important to understand which factors contribute to eating decisions and how we can intervene on them. Traditional psychological models have postulated that people reflect on behavioural options, form intentions and then translate these into behaviour. However, decades of research have shown that intentions are often not successfully enacted, a phenomenon termed the 'intention behaviour gap'.[3]

To investigate why people sometimes fail to transfer their intentions into behaviour, researchers have devised a range of indirect measures, typically assessed with computer tasks based on the measurement of reaction times (RTs),[4 5] as opposed to direct measures

Humans make several decisions per day about whether, what and how much to eat.[1] All these decisions have an influence on human

such as self-reported evaluations of stimuli. Under certain conditions, such indirect measures of food preference can increase prediction accuracy of actual behaviour, above and beyond questionnaire data[6 7] (but see[8] for a critical discussion). One example of such an RT task is the Approach Avoidance Task (AAT). In the AAT, participants usually use a joystick (or in more recent studies, a touch-screen[9 10]) to perform movements towards or away from different stimulus categories. These categories, such as foods and non-food objects, are compared on how fast they are approached and avoided, and this RT difference is termed the 'approach bias'.[11] Here, we adopt an operational definition, such that we define approach bias as the relative speed with which one can approach the target stimuli (eg, foods) in the AAT, remaining agnostic to what might be the underlying mental construct.[8 12 13] Food approach biases seem relevant to real world eating behaviour, as they have been found to be higher in people who strongly crave foods,[14] and they relate to increased food consumption in impulsive individuals and in people who are prone to external or emotional eating[15 16] (but see Spruyt *et al*[17] for contradictory findings from the alcohol domain and a wider discussion in Friese *et al* and Kakoschke *et al*[18 19]).

So far, it has mostly been ignored that approach biases may fluctuate over time. In most studies to date, approach bias has, at least implicitly, been treated as a relatively stable, trait-like phenomenon, in line with its conceptualisation as a (stable) mental construct. Approach bias is typically measured at a single time point and then correlated with other phenomena like trait food craving,[10] weight status,[20] or eating disorder diagnosis.[21] One recent study demonstrated that approach bias was independent of experimentally induced satiety, indicating stability of the bias across situations. However, participants' desire to eat specific foods did explain variance in approach bias, implying that bias might vary across time within individuals depending on their current consumption desires.[22] This is in line with the finding that approach bias for chocolate was positively correlated with current chocolate craving,[23–25] craving being an experience of intense desire for a specific food which is temporally variable by definition.[26] Other studies using a mobile version of the AAT indicated that test-retest reliability across eight measurement occasions was low while split-half reliability was high, again indicating temporal fluctuations in approach biases.[27] This is in line with findings obtained from other indirect measures that showed modest stability over time.[28] Such within-subject fluctuations in biases are probably not only due to random variation, as approach-avoidance biases have been shown to decrease with after-meal-satiety in normal-weight individuals, and they have been shown to change based on individuals' current affective states.[29–33] In combination, these results raise questions about the temporal and situational stability of approach biases.

Associations between behavioural approach bias and intake-related variables have led to the development of Approach-Avoidance Interventions (AAIs). Traditionally, it has been assumed that a stable mental construct thought to underlie the behavioural approach bias can be changed by repeatedly pairing unhealthy foods with avoidance and healthy foods (or neutral objects) with approach. This should then affect food intake. How exactly AAIs might work is, however, a matter of current debate. Based on the idea that the behavioural approach bias operationalises learnt associations between appetitive stimuli and approach, some authors argue that the repeated pairing of appetitive stimuli and avoidance weakens or reverses this association through the formation of new associations between stimuli, movement direction, and evaluative properties inherent to approach and avoidance.[23] Others, however, argue that stimulus value is updated due to a conflict between evaluation and within-task behaviour which then influences intake decisions,[34–36] or that altered behaviour is due to changed food evaluations, which are caused by cognitive inferences based on task requirements.[37 38] Independent of what might be the exact working mechanism, earlier studies have shown that the behavioural approach bias,[39] food choice,[40] and subsequent food intake[41] can be altered by AAIs. The evidence in this domain is mixed, however,[42–46] which might have several different reasons.

First, in some studies participants approach and avoid stimuli based on their category (eg, food vs objects) and in others, based on an irrelevant feature of the stimulus such as the frame colour or orientation. Such task differences may affect participants' awareness of the contingencies between stimulus and required response, as well as expectations about training effects. As this awareness could increase the effectiveness of the intervention, especially when AAIs change behaviour through cognitive inferences as noted above,[47] this study uses a relevant-feature AAT/AAI and closely tracks participants' contingency awareness. Second, the personal relevance of the trained stimuli may differ between interventions, and this may influence effectiveness.[48] Some interventions specifically try to retrain approach biases to chocolate in individuals reporting high trait-level chocolate craving or consumption,[39] while other interventions train responses to a preselected set of healthy and unhealthy foods without taking into account if participants actually consume the unhealthy foods or if healthy foods fit to individual needs of the participants (eg, in terms of taste, food intolerances). Third, most studies only deliver a single session of AAI (with Meule et al and Kakoschke *et al*[49 50] being exceptions), while more sessions might lead to larger effects that would be easier to detect.[51] Finally, the effectiveness of AAIs might depend on when the intervention is delivered. It is easy to see that interventions might be most fruitful in or just before moments when the risk for unhealthy intake is high.

Smartphone-based AATs and AAIs are interesting for a range of research questions that cannot be answered with stationary computer-based AATs and AAIs. First, smartphones allow easy delivery of AAI to participants during

their daily routines. This helps participants to perform the intervention repeatedly, and to bring the intervention temporally and spatially closer to 'high-risk' situations in everyday life. Assuming a rather fast decay of intervention effects, the closer proximity offered by the smartphone should enhance intervention effectiveness compared with conducting it on one's personal computer[52] or in a laboratory session.[53 54] Another advantage of repeated intervention through smartphones is the possibility to measure immediate and delayed intervention effects on fluctuating phenomena like food craving. Lastly, smartphones allow to measure bias more easily at any time of the day, and especially at moments when it may be relevant for food consumption. This allows us to examine the temporal and situational variability of approach bias. Combining it with repeated delivery of eating-related questions throughout the day (ie, ecological momentary assessment, EMA) also allows for correlating fluctuations in approach bias with other temporally variable phenomena like food craving, affect and intake.

Several studies have delivered interventions using computer tasks through the internet[52 53 55–57] and have generally reported good compliance rates and effects on dietary intake. Smartphone-based interventions using similar RT tasks are much rarer and have reported mixed results on key outcomes.[49–51 58] One of the two studies delivering smartphone-based AAI required participants to tilt the phone to respond, and found positive effects on food choice and approach bias towards unhealthy foods.[50] The other study found neither day-level nor longer-term effects of AAI using swipe movements, as compared with an EMA-only intervention and a sham training group.[49] It is important to note that that study did not find any approach bias in participants to begin with which suggests that the swipe movements did not clearly represent approach and avoidance and that the sample size was small. It is therefore unclear to what degree its results can be taken as evidence against the effectiveness of mobile AAI.

One recent AAT variant does not require swiping movements on the touchscreen, but instead requires participants to physically move the phone towards or away from themselves while viewing food stimuli.[27 59–61] This task has been shown to be a valid tool to measure food approach biases outside the laboratory and to provide relevant information beyond self-report measures.[59] In addition to RTs, it also yields data on the force of the movements, which might contain relevant information not captured by RTs.[61]

The study presented here sets out to test its effectiveness as an intervention tool for AAI; that is, when it is programmed to pair the foods that a specific participant wants to eat more often with approach, and to pair the foods that a specific participant wants to eat less often with avoidance responses. Specifically, we will study to what degree the intervention can support participants in their goal of changing their eating behaviour. We further examine the reliability and validity of approach bias scores

obtained through a phone-delivered AAT. Combining the AAI/AAT with the repeated measurement of related phenomena through EMA allows us to disentangle short-term and long-term intervention effects as well as to investigate whether approach bias covaries with intake-related variables over time.

## METHODS AND ANALYSIS
### Study overview
The study uses a two-arm, double-blind randomised controlled trial conducted with German-speaking participants, and is coordinated at the University of Salzburg, Austria. It compares an active AAI to a sham-training (a measurement Approach-Avoidance Task, AAT) in its impact on eating behaviour, food liking, food craving and food approach bias.

### Participants
Participants will be recruited via university e-mailing lists, social networks, university events and word of mouth. Participants must be between 18 years and 60 years of age, and must not be pregnant or report a diagnosis for an eating disorder. Importantly, participants must have an intention to change their eating behaviour (which they indicate on sign-up), without further specification regarding increased or decreased intake of certain foods or food categories.

To determine the required sample size, we performed a power analysis using pre-existing data from a (so far unpublished) study, where we attempted to change participants' approach-avoidance bias and analysed its change from pre-training to post-training (see online supplemental file 1 for more information). We opted to use this pre-existing data as the structure of the study is similar to the current study and the correlations and noise therein would be more likely to reflect the findings we will observe than would fully simulated data.

First, group-level differences between the pre-to-post effects of sham and active training were removed such that a time-by-group effect size of 0 was achieved. A new effect size was then applied by increasing the post-treatment group mean of the active training participants by a multiple of 16 between 64 and 244, giving effect sizes around $g=0.5$. After this, participants were randomly sampled with replacement such that sample sizes between 80 and 180 in multiples of 10 were achieved. Each combination of sample size and effect size was resampled and tested 200 times. After sampling a set of participants, a multilevel analysis was performed where approach-avoidance bias scores were predicted with fixed predictors of treatment group (sham or active) and time (pretraining or post-training), as well as random effects of time grouped by participant and time grouped by stimulus. The p value of the group by time interaction was recorded. The proportion of p values below .05 was computed to determine the power at each combination of sample size and effect size.

Based on this power analysis, we determined that a medium effect size ($g$=0.50) and a power of 0.80 would require a sample size of about 150 participants. Based on the effect size observed in that other study of $g$=0.56, 150 participants result in a power of about 0.88. A table with all power analysis outcomes is depicted in online supplemental file 2. With an estimated recruitment rate of three participants per week and allowing for recruitment difficulties slowing down the process, we expect data collection to last from November 2022 to roughly January 2024. Data collection continues until 150 participants are reached.

### Materials and procedure
#### Baseline questionnaires
In a web-based questionnaire (see online supplemental file 3), participants give informed consent (consent form see online supplemental file 4) and then indicate their age, gender, nationality, state of employment, highest achieved formal education level, dietary restrictions (vegan, vegetarian, pescetarian, omnivorous, other), height, weight, and possible food allergies or intolerances. Participants identifying as female or diverse are also asked about their menstrual cycle. Further, to assess exclusion criteria, they are asked if they are currently suffering from an eating disorder. This is followed by the stimulus selection (see below for details). Participants then complete the German versions of the following questionnaires: subscales restrained eating and external eating from the Dutch Eating Behaviour Questionnaire;[62]

the Salzburg Emotional Eating Scale;[63] the Salzburg Stress Eating Scale;[64] Perceived Self-Regulatory Success in Dieting;[65] the short version of the UPPS (Urgency; (lack of) Premeditation; (lack of) Perseverance; Sensation Seeking) Impulsivity Scale.[66]

#### Stimuli
We preselected 90 food and drink pictures from the food.pics[67] and CROCUFID[68] databases, and from freely available online resources based on typical availability in Austria and Germany as the main recruitment sites. At the beginning of the study, participants rated these 90 images on two scales: 'In the last three weeks, on how many days have you eaten/drunk this food/drink?' (recent intake) and 'In the next three weeks, on how many days would you like to eat/drink this food/drink?' (intended intake). We select the six foods with the most negative difference between past and intended consumption as 'increase foods' (eaten less often than intended). Then, among the foods that were eaten at least on 6 days in the past 3 weeks, we select six with the most positive difference between past and intended future consumption (eaten more often than intended = 'decrease-foods'). A randomly selected four of these six images are then used in the intervention phase in both groups while the other two were left untrained to test whether the intervention would be specific to the foods used in the task. A random selection of 8 out of a set of 12 images of office items serves as control stimuli. Figure 1 displays the selection of stimuli. The food stimuli were not categorised as 'healthy' or 'unhealthy', giving

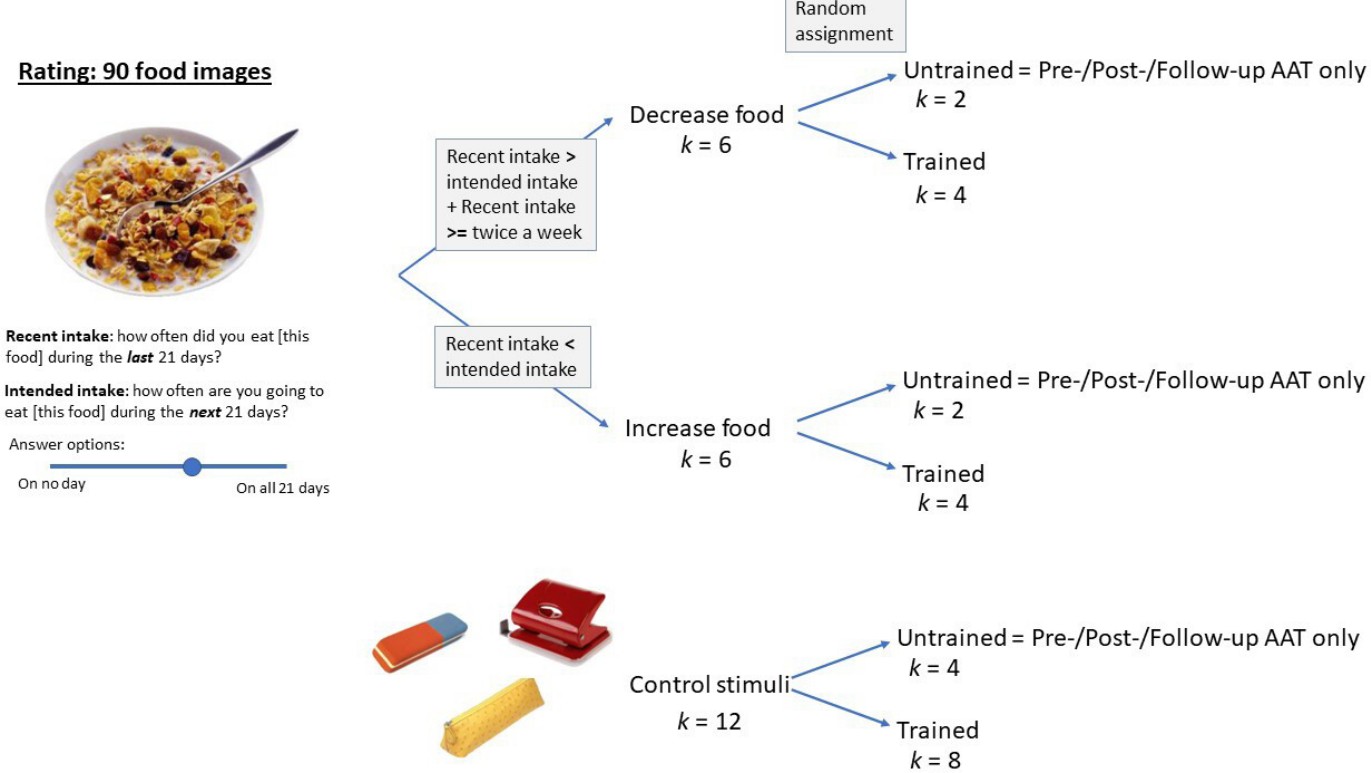

**Figure 1** Selection of personal food stimuli.

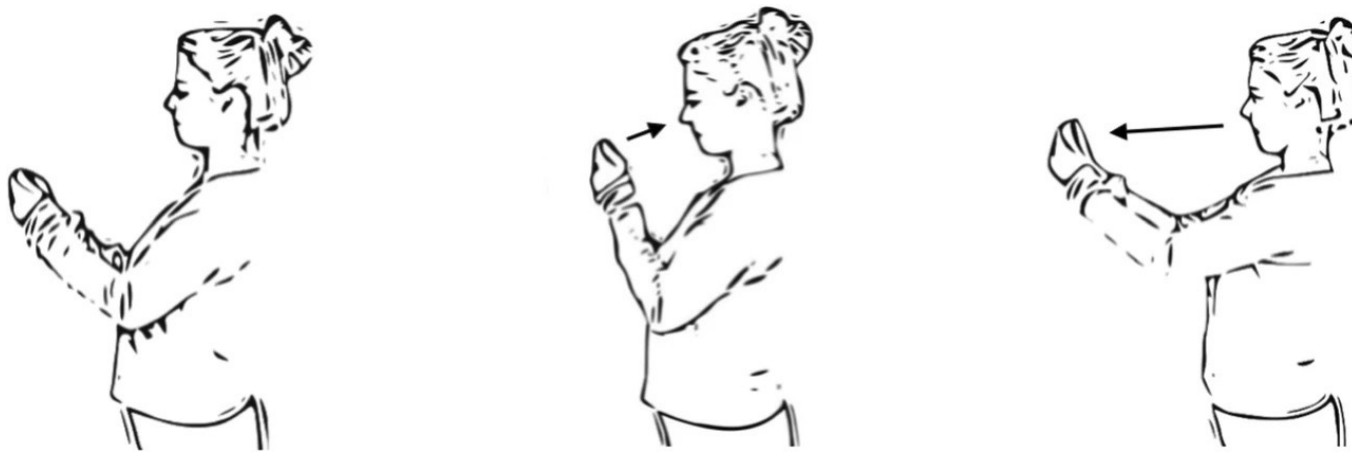

**Figure 2** Illustration of the task. On approach trials, participants move the phone closer to themselves (central image), on avoidance trials, they move it away from themselves (image on the right; adapted from Zech *et al* [61]).

participants full flexibility for choosing 'increase-foods' and 'decrease-foods'.

### Set-up call

Within a few days of filling out the online questionnaire, a member of the study team contacts participants to schedule a set-up call via phone or a videoconferencing tool. In this call, the member of the research team explains the procedure of the study. They further help participants install the necessary apps onto their smartphone (m-path KU Leuven, 2022; for EMA and the AAT app (The app can be downloaded on Android devices from this address: https://play.google.com/store/apps/details?id=com.eatlabsbg.eatapp)) and then confirm the correct selection of approach and avoidance foods as determined by the rating task. After the call, participants receive a manual for the study via email, which summarises the study procedures and the use of the smartphone applications and includes participants' individual three-digit identification code as well as contact information of the study team.

### Approach-Avoidance Task

We use the smartphone-based AAT as introduced by Zech *et al*.[61] In this version of the task, participants see stimuli on their horizontally held phone screen and they perform approach or avoidance movements by physically moving the phone towards/away from themselves (see figure 2 and two short introductory videos here: https://osf.io/4k3q9/?view_only=4db6431fd5ee4148a97f3be7f799ea4a). Each trial starts with a fixation dot in the middle of a white screen, which is followed by either one of the food or object stimuli after a 1500 ms delay. While correct approach or avoidance responses make the picture disappear and trigger the start of a new trial, incorrect responses are followed by a 2000 ms display of a black error-cross. If a participant does not respond for 2000 ms, a clock icon is displayed indicating timeout.

The active and sham AAT trainings feature 4 out of 6 approach-foods, 4 out of 6 avoid-foods, as well as 8 out of 12 control object stimuli. The training sessions consist of four training blocks of 16 trials each, and each training block is preceded by 4 practice trials, yielding a training session of 64 training trials and 16 practice trials, or 80 trials total. The pre-, post- and follow-up bias assessment AATs similarly consist of four blocks preceded each by four practice trials but feature all selected images (6 'increase-foods', 6 'decrease-foods' and 12 objects). All 24 images are presented one time per block, yielding 96 test trials and 16 practice trials, or 112 trials total. In all AATs, the instructions of the blocks alternate such that participants are instructed to approach foods while avoiding objects in the first block (approach-food blocks) and avoid foods while approaching objects in the second block (avoid-food blocks). This order is the same for all sessions and all participants. Crucially, in the active training AATs, only approach-foods are shown in the approach-food blocks, while only avoid-foods are shown in the avoid-food blocks; sham training instead features both approach-foods and avoid-foods during both approach-food and avoid-food blocks. Completing one session of the AAT/AAI takes about 5 min.

### Ecological momentary assessment

Participants follow the EMA schedule for a total of 20 days. During the whole period, participants receive two prompts per day (delivered through the smartphone application m-path[69]), one just before the time a participant usually eats lunch and the other in the evening (prompted at an individualised time agreed upon with the participant to represent an end-of-day signal). Table 1 shows the questions that participants answer on those prompts and figure 3 displays the temporal sequence of the study. EMA prompts on days 1–3 of the study only contain the listed questions. On day 4 (the day before the start of the intervention) and day 17 (the day after the end of the intervention), participants receive an instruction to open the AAT application and complete a measurement AAT. On every second day during the intervention phase (days 5 through 16), participants receive an instruction to open the AAT application and complete a training AAT

**Table 1** Overview of EMA questions (own translations from German)

| Item | Midday (all study phases) | Evening Days 1–4 and 17–20 | Evening Days 5–16 | Evening: follow-up |
|---|---|---|---|---|
| When was the last time you ate something? | X | | | |
| What type of meal was it? (breakfast, lunch, dinner, snack) | X | | | |
| How hungry are you right now? | X | X | X | |
| Do you feel like you have everything under control? | X | | | |
| Do you feel like you are on top of things? | X | | | |
| How optimistic do you feel right now? | X | | | X |
| How happy do you feel right now? | X | | | X |
| How lonely do you feel right now? | X | | | X |
| How depressed do you feel right now? | X | | | X |
| How angry do you feel right now? | X | | | X |
| How mad do you feel right now? | X | | | X |
| How tense do you feel right now? | X | | | X |
| How anxious do you feel right now? | X | | | X |
| How much do you want to stick to your dietary goals *for the rest of the day?* | X | | | |
| How much do you want to stick to your dietary goals *tomorrow?* | | X | | X |
| How strong has your craving for this food been *today?** | | X | | X |
| How much of this food have you eaten *today?** | | X | | X |
| Have you said things today without thinking? | X | X | X | X |
| Have you spent more money today than you wanted to? | X | X | X | X |
| Have you felt impatient today? | X | X | X | X |
| Have you made a spontaneous decision today? | X | X | X | X |
| How strong has your craving for this food been *since the midday questionnaire?** | | | X | |
| How much of this food have you eaten *since the midday questionnaire?** | | | X | |
| How much do you expect that this task will help you reach your dietary goals?† | X† | | | |
| Throughout the whole study, how often did you push this food away from yourself?* | | | | X |
| Please indicate the day your last period started. | | | | X |

All items are answered on a slider from 0 to 100 unless indicated differently in parentheses. Midday prompts remain the same across the entire study duration. Evening prompts differ depending on the study phase as indicated..
*These items are asked alongside a food image and are repeated for each image (six 'increase-food' and six 'decrease-food' images).
†This item is only asked after the first and the last AAT session (days 5 and 15).
AAT, Approach Avoidance Task; EMA, ecological momentary assessment .

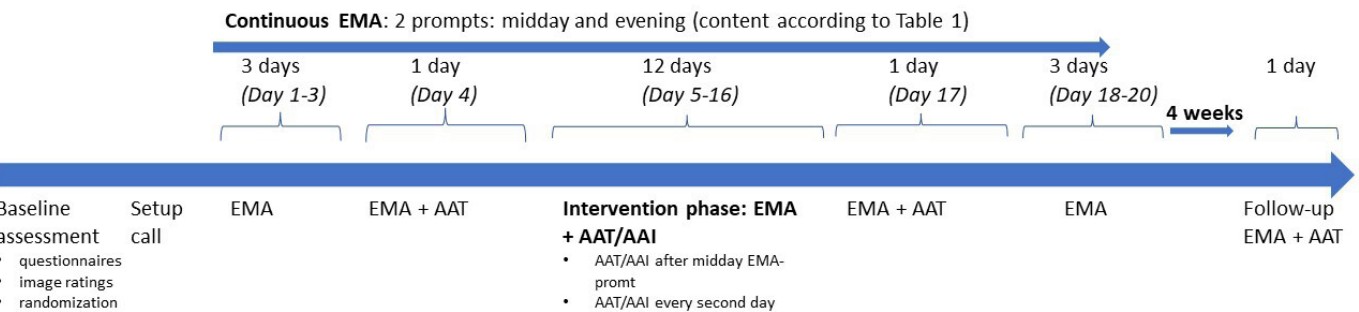

**Figure 3** Time schedule of the whole study period. AAI, approach-avoidance intervention; AAT, Approach Avoidance Task; EMA, ecological momentary assessment.

after completing the midday prompt. Thirty minutes after completing the midday prompt, participants receive a notification asking whether they conducted the training. On replying 'yes', they receive positive feedback; on replying 'no', they are asked to now open the AAT app to conduct the task. The number of sessions was chosen based on earlier, similar studies,[50 52] balancing participant burden, compliance and intervention intensity.

In addition, on day 6 and day 16 (the first day and the last day including an AAT/AAI session), participants further indicate their expectancy of how much the task will help them reach their dietary goals. Four weeks after the end of the initial 20-day EMA period, participants receive one additional EMA questionnaire and a measurement AAT in the evening. After performing this final AAT, participants indicate how often they believed they pushed or pulled each of their 'decrease-foods' and 'increase-foods'.

## Procedure

The procedure for study participation is as follows: after interested participants contact the study team, they receive an individual participant code and a web link to the baseline questionnaire. At this point, an R-script (The function *sample* randomly outputs the number '1' or '2' which correspond to the conditions.) randomises participants to either the intervention or control group with the condition unknown to the study team. After a set-up call with a member of the research team within a few days of filling out the questionnaires, participants start receiving EMA prompts and AAT as described above. Figure 3 shows the timeline of the whole study. Throughout the study period, participants can contact study personnel who also monitor compliance to the EMA and AAT schedule and contact participants in case of low compliance: participants receive an email if they miss more than one of the first three AAT/AAI sessions.

## Outcomes
### Main outcomes

This study uses three main outcome measures. The first outcome measure is participants' self-reported intake of 'increase' and 'decrease' foods according to the EMA schedule outlined above, on a slider from 0 (labelled 'nothing') to 100 ('very much'). The second outcome measure is participants' self-reported craving for those

same foods in the same manner. Simple, single-item measures reduce participant burden but might negatively affect reliability. To ameliorate this, measures are applied for each food separately. Time trends that might indicate changes in participants' perceptions of amounts as 'much' or 'little' will be checked in the control group and, if present, controlled for in analyses.

The third outcome measure is the approach bias for all selected foods based on the RT and force in the AAT. The RT is defined as the time from picture onset to movement onset. Force is defined as the peak acceleration in the correct direction during a trial, standardised within participant by dividing every individual's measurement of force by the participant-specific SD. Separately for approach and avoidance trials as well as for sessions, the RT and the force will be averaged across the four AAT blocks for each specific food stimulus. For objects, we will also average across the different stimuli. The average approach or avoid response for objects on a session will be subtracted from stimulus-specific food approach or avoidance response on that session to achieve food-specific *single-difference* approach and avoidance scores according to these formulas:

$$\text{Stimulus} - \text{specific avoidance} = [\text{food} - \text{specific avoidance}] - [\text{average object avoidance}]$$

$$\text{Stimulus} - \text{specific approach} = [\text{food} - \text{specific approach}] - [\text{average object approach}]$$

*Double-difference scores* will be used as a full bias score per food stimulus and session, according to the formula: ((food-specific avoidance)−(food-specific approach))−((average object avoidance)−(average object approach)).

### Secondary outcomes

Dietary intentions are measured according to the outlined EMA schedule.

### Data analysis plan

Data analysis will serve to investigate a series of research questions relating to different aspects of the study. In this section we will provide a brief description. Full details on the data analysis, including the exact multilevel analysis formulas, are available in the preregistration at https://osf.io/yn7kt .

## Data exclusion

For analyses regarding the effectiveness of the AAI, we exclude participants who did not conduct any of the AATs during the intervention phase, as we regard them as 'not treated'. For the remaining participants, sensitivity analyses are performed to test whether the number of completed training sessions affects intervention effectiveness. For analyses of approach bias, we exclude error trials and trials with RTs that deviate more than ±3 SD from the individual mean of the participant in that AAT session. If more than 25% of trials must be excluded based on these criteria, the whole AAT session is excluded from further analysis. This post hoc session exclusion does not affect whether a participant is counted as 'not treated' or not in the analyses regarding the effectiveness of the AAI.

## Overall intervention effectiveness

The first set of research questions relates to the effectiveness of the intervention as compared with the control condition from pretraining to post-training. To this end, we use multilevel models to predict intake of trained 'increase' and 'decrease' foods as a function of timepoint (3 days preintervention vs postintervention), condition (intervention vs control) and their interaction. Equivalent models test the intervention effect on approach biases towards—and craving for—the two food categories. To test to what degree the effect of the training intervention is specific to trained foods, we then use data from trained and untrained stimuli and add a variable that indicates whether a food appeared in the training or not (trained vs untrained) and all interaction terms to the model. This is followed up with tests to determine whether changes in the approach bias are mainly driven by changes in approach—or avoidance RTs. We further test the moderating role of intentions, baseline stimulus craving and person-level variables obtained from the questionnaires, as well as contingency awareness and expectancy by adding the relevant variable and its interaction terms to the equations. Finally, we examine the mediating effect of craving for changes in intake.

## Immediate intervention effectiveness

The second set of research questions concerns the short-term effects of the intervention during the intervention phase (days 5–16). Multilevel models predicting food intake and cravings, respectively, include the factors group (intervention vs control) and (off-)training day (training day vs no training day) as predictors. In another pair of multilevel models, we use group and the number of days since the beginning of the intervention and their interaction as predictors of craving and food intake, respectively. The force applied during the training is used as a predictor for the change in craving and intake from before the start of the training.

## Trait and state components of approach bias

The third set of research questions relates to the state and trait components of approach bias and is examined within the control group only. This is because only participants in the control group receive measurement AATs throughout the study period. Multilevel models test whether bias size and negative emotions are related on both a between-subjects and a within-subjects level and to what degree this depends on the strength of the desire for these foods. A separate model tests equivalent research questions for the relation between bias strength and craving, as well as bias strength and intake, respectively. The latter analyses testing how bias strength is related to subsequent food intake are expanded by including trait and day-level impulsivity and day-level intentions of regulating food intake.

## Patient and public involvement

Patients and the public are not involved in study design, data collection, data analysis, or dissemination.

## Ethics, dissemination and data handling

The study has received ethical approval from the ethics board of the University of Salzburg and is conducted in accordance with the declaration of Helsinki. Results of the trial will be disseminated through a series of articles in appropriate scientific journals and conference presentations.

Data are handled confidentially and stored in a pseudonymised manner. Neither m-path nor the AAT application collect personal data but work through three-digit identification codes assigned to participants. The identification key linking personal data to the identification codes will be kept in password-protected files separately from the pseudonymised data and will be destroyed 1 year after termination of the study. Deidentified data will be archived for at least 10 years and consent forms as documentation of participation will be archived for 30 years. The deidentified data will be made public on the Open Science Framework after the completion of planned publications.

**Contributors** MBA: conceptualisation, methodology, writing first draft of the protocol, revising the protocol. HvA: conceptualisation, methodology, analysis plan, reviewing drafts of the protocol. SK: conceptualisation, methodology, analysis plan, reviewing drafts of the protocol. JB: conceptualisation, methodology, supervision, reviewing drafts of the protocol. All authors read and approved the final version of the protocol.

**Funding** This research was funded in whole, or in part, by the Austrian Science Fund (FWF) [grant number P 34542-B]. SK and HvA were supported by the Doctoral College 'Imaging the Mind' (FWF; W1233-B). HvA was additionally supported by the project: Mapping neural mechanisms of appetitive behaviour (FWF; KLI762-B). The funder plays no role in the study design; collection, management, analysis, and interpretation of data; writing of the report; and the decision to submit the report for publication. For the purpose of open access, the author has applied a CC BY public copyright licence to any Author Accepted Manuscript version arising from this submission.

**Competing interests** None declared.

**Patient and public involvement** Patients and/or the public were not involved in the design, or conduct, or reporting, or dissemination plans of this research.

**Patient consent for publication** Not applicable.

**Provenance and peer review** Not commissioned; externally peer reviewed.

**ORCID iD**
Matthias Burkard Aulbach http://orcid.org/0000-0003-3830-2867

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
