## [Reviewer comments · BMJ Open]

ARTICLE DETAILS

TITLE (PROVISIONAL)	Study protocol for a Randomized-Controlled Trial to Test the Effectiveness of a Mobile Approach Avoidance Intervention and to Measure Approach Biases in an Ecological Momentary Assessment context
AUTHORS	Aulbach, Matthias; van Alebeek, Hannah; Kahveci, Sercan; Blechert, Jens

VERSION 1 – REVIEW

REVIEWER	Kakoschke, Naomi Commonwealth Scientific and Industrial Research Organisation (CSIRO), Nutrition and Health Program (Health & Biosecurity)
REVIEW RETURNED	20-Dec-2022

GENERAL COMMENTS	The authors presented a clear and well justified study protocol for an RCT examining the impact of mobile-delivered AAT for modifying food approach bias, self-reported intake and craving. I only have minor suggestions for improving the study description: -Are the foods participants select from within a category based on healthiness or caloric levels? Otherwise, participants may select to approach unhealthy foods and vice versa. Moreover, how were the set of 90 food images selected and were these matched on image and food characteristics? Were they classified as healthy and unhealthy or could participants select any foods to increase consumption of?-It would be useful to clarify how intention to change eating behaviour was defined e.g., to eat more healthily (general) or to reduce consumption of junk food (specific) etc.-Was an attrition rate also estimated? Was the recruitment rate based on previous studies? How many participants were targeted given an anticipated enrolled n=150?- Intervention: What was the rationale for using a relevant feature version of the AAT and the rationale for the study duration? It would also be useful to clarify how long the training takes to complete each day.-Is the second EMA questionnaire each day prompted at an individualised time for each participant or participant triggered?-Procedure: How was low compliance defined (e.g., 80%)?
---

REVIEWER	Van Dessel , Pieter Ghent University, Department of Experimental-Clinical and Health Psychology
REVIEW RETURNED	28-Dec-2022

GENERAL COMMENTS

This paper reports the study protocol for an RCT in which the authors will test the effectiveness of smartphone-based approach-avoidance training for (healthy) eating behavior. To this end, participants will be randomly assigned to complete six sessions of either approach-avoidance training or sham training. The approach-avoidance training will involve the repeated approach or avoidance of specific (self-selected) foods. Training effects will be assessed on self-reported food intake and craving (via ecological momentary assessment) as well as on behavioral performance in the approach-avoidance task.

Overall, this paper was an interesting read. The research question of whether smartphone-based approach-avoidance training can be effective for changing food intake is an important one both from a theoretical and practical point of view. The study protocol is also carefully designed (e.g., careful selection of the training task, self-selected stimuli, EMA assessment) and the authors are very thoughtful in their explanation of the research design, with sufficient detail (e.g., in terms of procedure and analyses). I also appreciate the authors' efforts to engage in good scientific practices such as pre-registration, making the data available on OSF, and publishing this study protocol.

That said, there are a number of things that came to mind while I was reading the paper that the authors might use to further improve the paper and the study protocol.

Most importantly, there is a lot of unclarity and inaccuracy in the description of the intervention. In the highlights section, the authors note that "The study is a randomized controlled trial testing an m-health intervention to assist participants in implementing their dietary intentions by re-training their automatic approach bias towards to-be-reduced foods...". This is inaccurate. The approach-avoidance training intervention does not directly 're-train' automatic approach bias. Whether the training changes automatic approach bias is a research question and testing this question requires that the authors indicate what they mean with "automatic approach bias". What is an approach bias? And to what extent is there any evidence that this bias is automatic?

The authors seem to take a couple of theoretical ideas for granted. For instance, in the introduction, they speak of impulsive processes that can be assessed with reaction-time tasks. It is unclear what this means and whether there is evidence for this idea. They also speak of indirect measures of food preference that can increase prediction of food intake. However, there is actually very little evidence for the predictive value of implicit measures above and beyond (well-controlled) self-report measures (e.g., see Van Dessel et al., 2021). The authors then move on to talk about the approach-avoidance task and indicate that a reaction time difference in this task indicates "the strength of an approach-avoidance bias". It is unclear, however, what the authors mean here. Do they see approach-avoidance bias as indicating food preference? Why? It is further explained that these biases "seems relevant to real-world eating behavior" as they for instance are higher in people who crave foods. However, there is also evidence that approach-avoidance biases often do not relate to real-life behavior or relate to it in an unexpected manner (e.g., see Spruyt et al., 2013). The authors then move on to talk about approach-avoidance training as an intervention to change this (still

	undefined) approach-avoidance bias as if it is an important (mental?) construct. I was happy to read the paragraph about the trait versus state-like phenomenon of approach bias but also in that paragraph I could not find how the authors define approach-avoidance bias. Do the authors consider it as an approach-avoidance tendency that is defined at the mental level? The authors refer to such tendencies in the Abstract but not in the text. In general then, it would be useful if the authors are much clearer about what they mean with approach-avoidance bias. I would suggest that they use this term to refer to a behavioral phenomenon, such as the faster response with an approach then with an avoidance response to certain stimuli. In that way, it is clear that a reaction time difference in an approach-avoidance task does not “indicate the strength of an approach-avoidance bias”. Instead, the RT difference IS the approach-avoidance bias. It can then also be explained that this bias is not stable and that there is not one specific task that can accurately identify this bias: The bias is simply a behavioral phenomenon that is observed in a given task. Next, it can then be explained whether the bias that is observed in a given task sometimes relates to (more interesting) behavior (such as real-life eating behavior). It can also be explained that some have argued that a bias in a specific task directly relates to a mental construct (automatic approach-avoidance tendency) and that this has led to the development of a training task to change this behavioral bias and in this way cause change also in real-life (unwanted) behavior such as unhealthy eating. It is important at that point that the authors do not simply take that idea for granted (as they seem to do now) but that they explain more recent evidence about approach-avoidance training effects and integrate recent theoretical ideas about approach-avoidance training effects in this explanation. Indeed, while there is some (albeit limited) evidence that approach-avoidance training can change real-life behavior, the idea that this is due to a change in behavioral bias is not well supported at all (change in the behavioral bias typically does not mediate change in real-life behavior). Moreover, the idea that a measured behavioral bias directly relates to a mental construct (i.e., the mapping of behaviour onto a mental level construct) as has often been done in implicit measures research is problematic (e.g., see De Houwer et al., 2013; Van Dessel et al., 2020). Given these insights, it does not seem very valuable to focus on change in approach-avoidance bias as a key outcome for the proposed study. When people consistently approach a certain stimulus in a task, it is only logical that their performance for this response will improve. This is simply the concept of behavior automatization via repeated practice. It is unlikely, however, that this change in behavioral bias causes the change in real-life behavior that is sometimes observed after approach-avoidance training. In fact, recent theories about training effects argue that training effects reflect the learning of inferences that people make when completing the training. For instance, after repeatedly avoiding alcoholic drinks, alcoholics may learn to infer that they
--	--

are able to and should avoid alcohol drinks in real-life as well (see Wiers et al., 2020).

Irrespective of whether one supports an inferential theory or another theory, however, it can be interesting to see if approach-avoidance training can influence real-life behavior. I would therefore focus on this question and even leave out the AAT assessment part (as it may reduce intervention effects and I'm not sure it can help answer any important question).

Note that, if the authors agree, and make this change, there is still an important flaw in the proposed study design that needs to be addressed. The number of participants that will be recruited was determined by looking at an approach-avoidance bias effect in an unreported study. The authors seem to have found a rather big effect on approach-avoidance bias. Such effects are to be expected (they simply indicate automatization of repeated behavior) but may be of little value and, most importantly, they are often much bigger than training effects on real-life behavior. Given that the latter effects are of key interest, it would be best to power the study for a smaller effect. As it is now, any observed effects on real-life behavior would be unreliable because the statistical power for finding small effects is too weak. This is even more problematic because there are so many statistical analyses. As a result, it is very likely that significant effects will be found that only represent statistical flukes.

Some other more minor issues and questions:

- According to inferential theories of approach-avoidance training effects it is important that participants have the intention to change behavior to foster intervention effects. While this rationale is not discussed in the current paper, participants are selected based on their intention to change behavior (which is a good thing). It was unclear to me, however, whether the training focuses on the goal to eat more healthily or simply the goal to change eating patterns. Can participants for instance also choose to approach unhealthy foods as foods they want to eat more? Also, participants are only included if they "have an intention to change their eating behaviour". What is meant here?

- The selected measure of eating behavior may be suboptimal. Participants self-report how often they ate a certain food on a scale from 0 (not) - 100 (very much). However, an effect of the intervention on this rating could indicate that participants simply re-considered whether the amount they ate a food is much rather than that participants actually ate the food less. It is also unclear how reliable this measure is.

- There are (too) many analyses and the specific research hypotheses are not explained in detail for every analysis. What are the confirmatory analyses? Are the hypotheses also pre-registered? I could not access the pre-registration.

References

De Houwer, J., Gawronski, B., & Barnes-Holmes, D. (2013). A functional-cognitive framework for attitude research. *European Review of Social Psychology*, 24(1), 252–287.

	Spruyt, A., De Houwer, J., Tibboel, H., Verschuere, B., Crombez, G., Verbanck, P., ... Noël, X. (2013). On the predictive validity of automatically activated approach/avoidance tendencies in abstaining alcohol-dependent patients. DRUG AND ALCOHOL DEPENDENCE, 127(1–3), 81–86 Van Dessel, P., Cummins, J., Hughes, S., Kasran, S., Cathelyn, F., & Moran, T. (2020). Reflecting on Twenty-Five Years of Research Using Implicit Measures: Recommendations for their Future Use. Social Cognition, 38, 223-242. Wiers, R.W., Van Dessel, P., & Kopetz, C. (2020). ABC-training: a new theory-based form of cognitive bias modification to foster automatization of alternative choices in the treatment of addiction and related disorders. Current Directions in Psychological Science, 29, 499-505.
--	---

VERSION 1 – AUTHOR RESPONSE

Reviewer: 1

Miss Naomi Kakoschke, Commonwealth Scientific and Industrial Research Organisation (CSIRO)

Comments to the Author:

The authors presented a clear and well justified study protocol for an RCT examining the impact of mobile-delivered AAT for modifying food approach bias, self-reported intake and craving. I only have minor suggestions for improving the study description:

-Are the foods participants select from within a category based on healthiness or caloric levels? Otherwise, participants may select to approach unhealthy foods and vice versa. Moreover, how were the set of 90 food images selected and were these matched on image and food characteristics? Were they classified as healthy and unhealthy or could participants select any foods to increase consumption of?

We refrained from imposing an a-priori distinction of “healthy” and “unhealthy” foods and rather let participants decide freely which foods they want to eat more/less of. While we are aware of the risk that this could potentially lead to an overall less healthy consumption pattern, we want to test Approach-Avoidance interventions as support for participants to more successfully implement their dietary intentions, regardless of their content. We have now explicitly stated that in the manuscript. We have further clarified that we pre-selected the 90 images based on typical availability in Austria and Germany as the main recruitment sites.

“We pre-selected 90 food and drink pictures from the food.pics [67] and CROCUFID [68] databases, and from freely available online resources based on typical availability in Austria and Germany as the main recruitment sites. At the beginning of the study, participants rated these 90 images on two scales: “In the last three weeks, on how many days have you eaten/drunk this food/drink?” (recent intake) and “In the next three weeks, on how many days would you like to eat/drink this food/drink?” (intended intake).”

“The food stimuli were not categorized as “healthy” or “unhealthy”, giving participants full flexibility for choosing “increase-foods” and “decrease-foods”.”

-It would be useful to clarify how intention to change eating behaviour was defined e.g., to eat more healthily (general) or to reduce consumption of junk food (specific) etc.

This is a good point and we clarified in the “Participants” section that there was no further specification regarding the intention to change eating behavior.

“Importantly, participants must have an intention to change their eating behavior (which they indicate upon sign-up), without further specification regarding increased or decreased intake of certain foods or food categories.”

-Was an attrition rate also estimated? Was the recruitment rate based on previous studies? How many participants were targeted given an anticipated enrolled n=150?

We did not estimate an attrition rate but have specified that we will recruit 150 valid participants, that is, participants that have completed at least one session of AAT/AAI. Since the trial is still ongoing, we cannot say how many participants will have been targeted by the end of the study.

- Intervention: What was the rationale for using a relevant feature version of the AAT and the rationale for the study duration?

The introduction section already discusses the issue of relevant vs irrelevant feature AATs but we have now added that this is the rationale for us to use a relevant feature version.

“Firstly, in some studies participants approach and avoid stimuli based on their category (e.g., food vs objects) and in others, based on an irrelevant feature of the stimulus such as the frame color or orientation. Such task differences may affect participants’ awareness of the contingencies between stimulus and required response, as well as expectations about training effects. As this awareness could increase the effectiveness of the intervention, especially when AATs change behavior through cognitive inferences as noted above [47], this study uses a relevant-feature AAT/AAI and closely tracks participants’ contingency awareness”

Regarding the study duration, we have added information to the methods section:

“The number of sessions was chosen based on earlier, similar studies [50,52], balancing participant burden, compliance, and intervention intensity.”

It would also be useful to clarify how long the training takes to complete each day.

We have added this important information to the Methods section.

“Completing one session of the AAT/AAI takes about five minutes.”

-Is the second EMA questionnaire each day prompted at an individualised time for each participant or participant triggered?

We have clarified that this interaction is prompted in the Methods section.

“Participants follow the EMA schedule for a total of 20 days. During the whole period, participants receive two prompts per day (delivered through the smartphone application m-path [69]), one just before the time a participant usually eats lunch and the other in the evening (prompted at an individualized time agreed-upon with the participant to represent an end-of-day signal).

-Procedure: How was low compliance defined (e.g., 80%)?

“Throughout the study period, participants can contact study personnel who also monitor compliance to the EMA and AAT schedule and contact participants in case of low compliance: participants receive an e-mail if they miss more than one of the first three AAT/AAI sessions.”

Reviewer: 2

Dr. Pieter Van Dessel , Ghent University

Comments to the Author:

This paper reports the study protocol for an RCT in which the authors will test the effectiveness of smartphone-based approach-avoidance training for (healthy) eating behavior. To this end, participants will be randomly assigned to complete six sessions of either approach-avoidance training or sham training. The approach-avoidance training will involve the repeated approach or avoidance of specific (self-selected) foods. Training effects will be assessed on self-reported food intake and

craving (via ecological momentary assessment) as well as on behavioral performance in the approach-avoidance task.

Overall, this paper was an interesting read. The research question of whether smartphone-based approach-avoidance training can be effective for changing food intake is an important one both from a theoretical and practical point of view. The study protocol is also carefully designed (e.g., careful selection of the training task, self-selected stimuli, EMA assessment) and the authors are very thoughtful in their explanation of the research design, with sufficient detail (e.g., in terms of procedure and analyses). I also appreciate the authors' efforts to engage in good scientific practices such as pre-registration, making the data available on OSF, and publishing this study protocol. We would like to thank the reviewer for these positive comments!

That said, there are a number of things that came to mind while I was reading the paper that the authors might use to further improve the paper and the study protocol.

Most importantly, there is a lot of unclarity and inaccuracy in the description of the intervention. In the highlights section, the authors note that "The study is a randomized controlled trial testing an m-health intervention to assist participants in implementing their dietary intentions by re-training their automatic approach bias towards to-be-reduced foods...". This is inaccurate. The approach-avoidance training intervention does not directly 're-train' automatic approach bias. Whether the training changes automatic approach bias is a research question and testing this question requires that the authors indicate what they mean with "automatic approach bias". What is an approach bias? And to what extent is there any evidence that this bias is automatic?

We agree with the reviewer that this wording takes the proposed mechanism as a given and we have changed this section accordingly. Of course, this connects to the issues elaborated on below so we respond to those theoretical matters in our responses to the following paragraphs.

The authors seem to take a couple of theoretical ideas for granted. For instance, in the introduction, they speak of impulsive processes that can be assessed with reaction-time tasks. It is unclear what this means and whether there is evidence for this idea. They also speak of indirect measures of food preference that can increase prediction of food intake. However, there is actually very little evidence for the predictive value of implicit measures above and beyond (well-controlled) self-report measures (e.g., see Van Dessel et al., 2021). The authors then move on to talk about the approach-avoidance task and indicate that a reaction time difference in this task indicates "the strength of an approach-avoidance bias". It is unclear, however, what the authors mean here. Do they see approach-avoidance bias as indicating food preference? Why? It is further explained that these biases "seems relevant to real-world eating behavior" as they for instance are higher in people who crave foods. However, there is also evidence that approach-avoidance biases often do not relate to real-life behavior or relate to it in an unexpected manner (e.g., see Spruyt et al., 2013). The authors then move on to talk about approach-avoidance training as an intervention to change this (still undefined) approach-avoidance bias as if it is an important (mental?) construct.

I was happy to read the paragraph about the trait versus state-like phenomenon of approach bias but also in that paragraph I could not find how the authors define approach-avoidance bias. Do the authors consider it as an approach-avoidance tendency that is defined at the mental level? The authors refer to such tendencies in the Abstract but not in the text.

In general then, it would be useful if the authors are much clearer about what they mean with approach-avoidance bias. I would suggest that they use this term to refer to a behavioral phenomenon, such as the faster response with an approach then with an avoidance response to certain stimuli. In that way, it is clear that a reaction time difference in an approach-avoidance task does not "indicate the strength of an approach-avoidance bias". Instead, the RT difference IS the

approach-avoidance bias. It can then also be explained that this bias is not stable and that there is not one specific task that can accurately identify this bias: The bias is simply a behavioral phenomenon that is observed in a given task. Next, it can then be explained whether the bias that is observed in a given task sometimes relates to (more interesting) behavior (such as real-life eating behavior). It can also be explained that some have argued that a bias in a specific task directly relates to a mental construct (automatic approach-avoidance tendency) and that this has led to the development of a training task to change this behavioral bias and in this way cause change also in real-life (unwanted) behavior such as unhealthy eating.

It is important at that point that the authors do not simply take that idea for granted (as they seem to do now) but that they explain more recent evidence about approach-avoidance training effects and integrate recent theoretical ideas about approach-avoidance training effects in this explanation. Indeed, while there is some (albeit limited) evidence that approach-avoidance training can change real-life behavior, the idea that this is due to a change in behavioral bias is not well supported at all (change in the behavioral bias typically does not mediate change in real-life behavior).

Moreover, the idea that a measured behavioral bias directly relates to a mental construct (i.e., the mapping of behaviour onto a mental level construct) as has often been done in implicit measures research is problematic (e.g., see De Houwer et al., 2013; Van Dessel et al., 2020).

We want to thank the reviewer for these detailed theoretical considerations on the issue and agree that the initial version of the introduction remained vague in terms of the conceptualization of approach bias. The revised text now mentions different ways of interpreting approach bias while remaining agnostic about its nature as a mental construct. In fact, one of the goals of the current study is to further our understanding how and to what degree approach bias relates to other constructs, as outlined in the section on the temporal variability of approach biases.

Seeing that the abstract had made references to automatic approach tendencies we understand the concern around different theoretical approaches to AAI effects. We have therefore revised those parts that were ambivalent regarding those points. Given the tight space constraints, however, we cannot go into a deep discussion about the precise mechanisms (association formation vs inferential account) involved in approach-avoidance interventions. We have added a short section about this issue in the introduction:

“Traditionally, it has been assumed that a stable mental construct thought to underlie the behavioral approach bias, can be changed by repeatedly pairing unhealthy foods with avoidance and healthy foods (or neutral objects) with approach. This should then affect food intake. How exactly AAIs might work is however a matter of current debate. Based on the idea that the behavioral approach bias operationalizes learned associations between appetitive stimuli and approach, some authors argue that the repeated pairing of appetitive stimuli and avoidance weakens or reverses this association through the formation of new associations between stimuli, movement direction, and evaluative properties inherent to approach and avoidance [23]. Others however argue that stimulus value is updated due to a conflict between evaluation and within-task behavior which then influences intake decisions [34–36], or that altered behavior is due to changed food evaluations, which are caused by cognitive inferences based on task requirements [37,38].”

Given these insights, it does not seem very valuable to focus on change in approach-avoidance bias as a key outcome for the proposed study. When people consistently approach a certain stimulus in a task, it is only logical that their performance for this response will improve. This is simply the concept of behavior automatization via repeated practice. It is unlikely, however, that this change in behavioral bias causes the change in real-life behavior that is sometimes observed after approach-avoidance training. In fact, recent theories about training effects argue that training effects reflect the learning of inferences that people make when completing the training. For instance, after repeatedly avoiding alcoholic drinks, alcoholics may learn to infer that they are able to and should avoid alcohol drinks in real-life as well (see Wiers et al., 2020).

While we agree that taking the idea of approach bias representing “implicit evaluation” (or any other one-to-one matching of approach bias and a mental construct) at face value is problematic, we do think that measuring approach bias and its change over the intervention period is valuable. While it might be unlikely that changes in approach bias mediate intervention effects on eating behavior, this remains an empirical question. Should we find that average approach bias and food intake change but that those changes are not related, this speaks for the idea that changes in approach biases are simply a result of motoric learning without relevance for the more relevant behavior (food intake). This would increase the awareness of the field about viewing bias changes and behavioral changes as two separate things. Should these changes be associated, however, this would be an argument that approach bias change is relevant to changes in dietary behavior. That is not to say that we can pinpoint the mechanisms underlying such mediation with the given set of measures. As you say: Inferences might change, but also other mechanisms could be responsible: behavior-stimulus interaction theory (Veling et al., 2008) for example, holds that avoiding positively valued stimuli is difficult and a devaluation of that stimulus will make avoiding easier. That is: it is easier for participants to get through the task quickly and with little errors. Such stimulus changes would then show up on our bias assessment post-training. Please also note that our feature relevant setup makes the association of stimulus class and response quite obvious, a fact that we try to tap into with our contingency awareness measures. While this means that we have to soften our claims in automaticity (in terms of non-intentionality) this coupling of movement direction and stimulus class could lead to a number of mental processes, be it inferences, social desirability, stimulus value changes or other. Further research would require an empirical demonstration, which is what we aim to do here.

Irrespective of whether one supports an inferential theory or another theory, however, it can be interesting to see if approach-avoidance training can influence real-life behavior. I would therefore focus on this question and even leave out the AAT assessment part (as it may reduce intervention effects and I’m not sure it can help answer any important question).

As we outlined in the above paragraph, we respectfully disagree with the suggestion of omitting the AAT assessments. We would also like to point out that changes to the study design are not possible at this stage as data collection is ongoing as per BMJ Open’s policy on protocol papers (“The intention of peer review is not to alter the study design.”, see <https://bmjopen.bmj.com/pages/authors/>)

Note that, if the authors agree, and make this change, there is still an important flaw in the proposed study design that needs to be addressed. The number of participants that will be recruited was determined by looking at an approach-avoidance bias effect in an unreported study. The authors seem to have found a rather big effect on approach-avoidance bias. Such effects are to be expected (they simply indicate automatization of repeated behavior) but may be of little value and, most importantly, they are often much bigger than training effects on real-life behavior. Given that the latter effects are of key interest, it would be best to power the study for a smaller effect. As it is now, any observed effects on real-life behavior would be unreliable because the statistical power for finding small effects is too weak. This is even more problematic because there are so many statistical analyses. As a result, it is very likely that significant effects will be found that only represent statistical flukes.

Again, a change in the study design is not feasible at this stage and we are bound to our preregistration in the German clinical trials registry (registration number DRKS00030780). We have, however, provided more detail on the power analysis that also allows determining how high power is when assuming smaller effect sizes (see Supplementary Table 1 and the Methods section).

Some other more minor issues and questions:

- According to inferential theories of approach-avoidance training effects it is important that participants have the intention to change behavior to foster intervention effects. While this rationale is not discussed in the current paper, participants are selected based on their intention to change

behavior (which is a good thing). It was unclear to me, however, whether the training focuses on the goal to eat more healthily or simply the goal to change eating patterns. Can participants for instance also choose to approach unhealthy foods as foods they want to eat more? Also, participants are only included if they “have an intention to change their eating behaviour”. What is meant here?

This was indeed unclear in the original version and we have now clarified that any type of intention to change dietary behavior was eligible and that foods were not a priori specified as “healthy” or “unhealthy”.

“Importantly, participants must have an intention to change their eating behavior (which they indicate upon sign-up), without further specification regarding increased or decreased intake of certain foods or food categories.”

“The food stimuli were not categorized as “healthy” or “unhealthy”, giving participants full flexibility for choosing “increase-foods” and “decrease-foods”.”

- The selected measure of eating behavior may be suboptimal. Participants self-report how often they ate a certain food on a scale from 0 (not) - 100 (very much). However, an effect of the intervention on this rating could indicate that participants simply re-considered whether the amount they ate a food is much rather than that participants actually ate the food less. It is also unclear how reliable this measure is.

While we agree that this measure is far from optimal, measuring food intake in daily life is notoriously difficult and unreliable (Thompson & Subar, 2017). We therefore decided to use simple measures to reduce participant burden and to increase its reliability by applying the measure for each food separately. It would be surprising if participants would change their perception of how much is “much” of a certain food. If such trends were present in the control condition we could correct for such time trends in the analyses.

- There are (too) many analyses and the specific research hypotheses are not explained in detail for every analysis. What are the confirmatory analyses? Are the hypotheses also pre-registered? I could not access the pre-registration.

We apologize that the preregistration was not visible, this should now be fixed: <https://osf.io/yn7kt>.

The preregistration provides more detail on the hypotheses and how they will be tested. Due to space constraints, it was not possible to include all of them in the current manuscript.

While we agree that there are many analyses, we considered it better to preregister them than to not mention them and then exploring these (in our view interesting and relevant) questions as follow-up analyses after data collection.

References

De Houwer, J., Gawronski, B., & Barnes-Holmes, D. (2013). A functional-cognitive framework for attitude research. *European Review of Social Psychology*, 24(1), 252–287.

Spruyt, A., De Houwer, J., Tibboel, H., Verschuere, B., Crombez, G., Verbanck, P., ... Noël, X. (2013). On the predictive validity of automatically activated approach/avoidance tendencies in abstaining alcohol-dependent patients. *DRUG AND ALCOHOL DEPENDENCE*, 127(1–3), 81–86

Van Dessel, P., Cummins, J., Hughes, S., Kasran, S., Cathelyn, F., & Moran, T. (2020). Reflecting on Twenty-Five Years of Research Using Implicit Measures: Recommendations for their Future Use. *Social Cognition*, 38, 223-242.

Wiers, R.W., Van Dessel, P., & Kopetz, C. (2020). ABC-training: a new theory-based form of cognitive bias modification to foster automatization of alternative choices in the treatment of addiction and related disorders. *Current Directions in Psychological Science*, 29, 499-505.

References (reply):

Thompson, F. E., & Subar, A. F. (2017). Chapter 1—Dietary Assessment Methodology. In A. M. Coulston, C. J. Boushey, M. G. Ferruzzi, & L. M. Delahanty (Eds.), *Nutrition in the Prevention and Treatment of Disease (Fourth Edition)* (pp. 5–48). Academic Press. <https://doi.org/10.1016/B978-0-12-802928-2.00001-1>

Veling, H., Holland, R. W., & van Knippenberg, A. (2008). When approach motivation and behavioral inhibition collide: Behavior regulation through stimulus devaluation. *Journal of Experimental Social Psychology*, 44(4), 1013–1019. <https://doi.org/10.1016/j.jesp.2008.03.004>

VERSION 2 – REVIEW

REVIEWER	Kakoschke, Naomi Commonwealth Scientific and Industrial Research Organisation (CSIRO), Nutrition and Health Program (Health & Biosecurity)
REVIEW RETURNED	08-Mar-2023
GENERAL COMMENTS	I thank the authors for their careful consideration of all editor and reviewer comments. The revised manuscript clearly addresses all comments. I look forward to seeing this manuscript published and to finding out the results of this important trial on mobile approach-avoidance training in the eating domain.

VERSION 2 – AUTHOR RESPONSE

We thank the reviewer for her kind comments and will be happy to provide the results of this trial as soon as possible.